# Development and Validation of a One-Step Reverse Transcription Real-Time PCR Assay for Simultaneous Detection and Identification of Tomato Mottle Mosaic Virus and Tomato Brown Rugose Fruit Virus

**DOI:** 10.3390/plants11040489

**Published:** 2022-02-11

**Authors:** Antonio Tiberini, Ariana Manglli, Anna Taglienti, Ana Vučurović, Jakob Brodarič, Luca Ferretti, Marta Luigi, Andrea Gentili, Nataša Mehle

**Affiliations:** 1CREA—Research Centre for Plant Protection and Certification, Via C.G. Bertero 22, 00156 Roma, Italy; ariana.manglli@crea.gov.it (A.M.); anna.taglienti@crea.gov.it (A.T.); luca.ferretti@crea.gov.it (L.F.); marta.luigi@crea.gov.it (M.L.); andrea.gentili@crea.gov.it (A.G.); 2Department of Biotechnology and Systems Biology, National Institute of Biology, Večna pot 111, SI-1000 Ljubljana, Slovenia; ana.vucurovic@nib.si (A.V.); jakob.brodaric@nib.si (J.B.); natasa.mehle@nib.si (N.M.); 3School for Viticulture and Enology, University of Nova Gorica, Dvorec Lanthieri, Glavni trg 8, SI-5271 Vipava, Slovenia

**Keywords:** ToBRFV, ToMMV, *Solanum lycopersicum*, *Capsicum annum*, leaves detection, seeds detections, performance criteria

## Abstract

*Tobamovirus* species represent a threat to solanaceous crops worldwide, due to their extreme stability and because they are seed borne. In particular, recent outbreaks of tomato brown rugose fruit virus in tomato and pepper crops led to the establishment of prompt control measures, and the need for reliable diagnosis was urged. Another member of the genus, tomato mottle mosaic virus, has recently gained attention due to reports in different continents and its common features with tomato brown rugose fruit virus. In this study, a new real-time RT-PCR detection system was developed for tomato brown rugose fruit virus and tomato mottle mosaic virus on tomato leaves and seeds using TaqMan chemistry. This test was designed to detect tomato mottle mosaic virus by amplifying the movement protein gene in a duplex assay with the tomato brown rugose fruit virus target on the CP-3’NTR region, which was previously validated as a single assay. The performance of this test was evaluated, displaying analytical sensitivity 10^−5^–10^−6^-fold dilution for seeds and leaves, respectively, and good analytical specificity, repeatability, and reproducibility. Using the newly developed and validated test, tomato brown rugose fruit virus detection was 100% concordant with previously performed analyses on 106 official samples collected in 2021 from different continents.

## 1. Introduction

Tomato (*Solanum lycopersicum* L.) is one of the most important crops worldwide, with an annual production of around 180 million tonnes of fresh weight [1]. Viral diseases are major causes of yield losses in tomato production, and the more representative and important species that can seriously affect this crop belong to *Crinivirus*, *Cucumovirus*, *Begomovirus*, *Potexvirus*, *Potyvirus*, *Tospovirus*, *Polerovirus*, and *Tobamovirus* [2,3]. The latter genus has long been considered a threat for agricultural crops; tobacco mosaic virus (TMV) and tomato mosaic virus (ToMV) have been reported to infect tomato and other important solanaceous crops for more than a century, representing the type species of the genus, and the most noticeable and economically important viruses up to the development of resistance varieties [4,5]. Tobamoviruses are characterized to have stable virions with a mechanical transmission pathway. In most cases, virions contaminate the seed coat and infectivity is preserved in seeds for up to several years. In nurseries, during plantlets’ emersion, the tobamovirus on the seed coat may be transmitted to wounded roots [4]. Furthermore, although a low seed-to-seedling transmission rate is reported, due to the large-scale usage of seeds and seedlings in farming, the contribution of a single infected seedling to an outbreak, or long-distance transmission, may become significant [5]. Due to mechanical transmission, the primary source can be easily spread by contacts, hands, tools, the greenhouse structure, and bumblebees [6]. Tobamovirus infectious particles are reported to be stable in plant debris and in contaminated soil for years [7]. In recent years, a new member of the *Tobamovirus* genus, tomato brown rugose fruit virus (ToBRFV), was characterized and reported to induce symptoms of mosaic, discoloration, and deformation on leaves, and size reduction, discoloration, brown rugosity, and malformation on fruits, making them unmarketable; reduction in plant vigor was also observed in diseased plants. Such severe symptomatology was also observed in tomato cultivars harboring resistance genes (*Tm-1*, *Tm-2*, *Tm-2^2^*) against tobamoviruses [8]. After emerging in Israel and Jordan in 2014 [8,9], the virus was reported in several European countries [10] (e.g., Italy [11], United Kingdom [12], the Netherlands [13], Greece [14], and Spain [15]). In North America, ToBRFV was first reported in Mexico in 2018 [16], then it also spread in California [17]. In Asia, it was reported first in Turkey [18] and then in China in 2019 [19].

Due to its high transmissibility and resistance breaking, ToBRFV was included in the A2 list of the European and Mediterranean Plant Protection Organization (EPPO) and subjected to prompt measures (Commission Implementing Decision (EU) 2020/1191 and subsequent amendments) to prevent its spread and establishment in the European Union. Validated molecular tests for the detection and identification of ToBRFV are available in EPPO Standard PM 7/146 (1) [20]; for seed testing, only real-time reverse-transcription polymerase chain reaction (real-time RT-PCR) tests are recommended.

A member of subgroup I of tobamoviruses, which also includes TMV, ToMV, and ToBRFV, is tomato mottle mosaic virus (ToMMV). It was first reported in Mexico in 2013, on a sample collected in 2009 [21]; the associated disease is characterized by symptoms of plant stunting, severe mosaic, mottling, distortion, and necrosis on leaves, and fruit necrosis in tomato [21,22,23]; in pepper, symptoms such as apical yellowing and necrosis were observed [22,24]. After the first report, ToMMV has been detected in the United States [25], Spain [24], Israel [26], China [27], Brazil [28], and Czech Republic [29]. Pepper seed lots imported into Australia were also found to be infected by ToMMV [30]. Moreover, Mut Bertomeo [31] reported that the presence of ToMMV was confirmed in batches of tomato seeds imported by Spanish companies between 2019 and 2021. In their study, 130 tomato samples and 85 pepper samples from different countries were analyzed by end point RT-PCR, and ToMMV was detected in five tomato seed samples from China (three), the USA (one), and Israel (one).

In addition, ToMMV, which shares a close phylogenetic relationship with ToBRFV [32], has proven to partially break resistance of tomato resistant to ToMV [23]. According to Nagai et al. [33], the *Tm-2^2^* gene is involved in the resistance against ToMMV; this study reported that three of seven tomato varieties and hybrid lines experimentally inoculated with ToMMV tested positive for the virus, suggesting a hypothetical ability to overcome resistance genes [33]. Additional resistance tests were conducted on a certified ToMV-resistant variety by Sui et al.; around 10% of plants were successfully infected with ToMMV [23].

In view of the above, the EPPO Panel on Phytosanitary Measures recommended the inclusion of ToMMV in the EPPO Alert List (2020); Australia has implemented emergency measures for ToMMV, requiring testing for imported tomato and pepper seed lots; finally, express Pest Risk Analyses were recently performed in Germany [34] and the Netherlands [35]. In a context of envisaged phytosanitary measures against ToMMV, proper diagnostic tests for its detection and identification, as stated by EPPO PM 7/76 (5) [36], are critical for the appropriate application of such measures. In particular, because of the high risk of cross-reactivity that serological methods present in the detection of tobamoviruses, molecular methods represent the gold standard for the specific detection of viral species within this genus. To date, there are few published protocols for the detection and identification of ToMMV. These are Sanger sequencing of generic tobamovirus RT-PCR or nested RT-PCR products (e.g., [8,37]) and multiplex endpoint RT-PCR for TMV, ToMV, and ToMMV [23].

In view of the above, a real-time PCR-based method suitable for testing ToMMV in leaf and seed matrices is needed, as learned from ToBRFV (see the guidelines for seed testing of Commission Implementing Decision (EU) 2020/1191 and subsequent amendments). In this work, we report the development and validation, according to EPPO PM 7/98 (4) guidelines [38], of a duplex real-time RT-PCR assay for simultaneous detection and identification of ToMMV and ToBRFV in leaves and seeds. In addition, using this developed method, we checked some official samples already tested for the presence of ToBFRV to evaluate the presence of ToMMV, whose presence in commercial seeds could be underestimated, as reported by Mut Bertomeo [31].

To the best of our knowledge, this is the first molecular method developed for the simultaneous detection and identification of the two emerging viruses mentioned above, which currently pose the greatest threat to solanaceous crops worldwide.

## 2. Results

### 2.1. Primers/Probe Design and Evaluation

ToMMV-specific primers/probe sets obtained targeting coat protein (CP) and movement protein (MP) regions are summarized in Table 1. Sets 1 and 2 were labeled in HEX whereas sets 3 and 4 were labeled in Texas Red.

Preliminary reactions performed, including the ToMMV-positive/ToBRFV-positive/negative controls, showed that all the ToMMV primers/probe sets were specific and allowed correct identification of ToMMV, with good Cq values ranging from 6.8 to 13.6, and discriminated it from ToBRFV without cross-reactions (Figure 1). The best annealing temperature was set at 60 °C with primer and probe concentrations at 0.3 and 0.2 μM, respectively. For all the sets, no amplification signals were obtained for negative controls. The primer sets 1 and 2, targeting the CP region and labeled in HEX, showed the lowest level of relative fluorescence units (RPUs), with a value of 1000 and 500, respectively, whereas sets 3 and 4 showed RPU values above 1500 (Figure 1).

In view of these results, primer/probe set 2, targeting CP region and showing the lowest level of RPU, was excluded from further evaluation. The remaining primer/probe sets 1, 3, and 4 were tested in duplex/triplex assay with ToBRFV assays. Considering the targeting region/labeling matches, a total of four combinations were obtained and tested as reported in Table 2.

The results of duplex/triplex assays showed expected amplification curves with Cq values ranging from 8.8 to 19.2 for leaf samples and from 25.8 to 28.5 for seed samples (Table 2). All the duplex/triplex assays were able to discriminate ToMMV from ToBRFV infected samples and to detect both targets in the mixed infected sample. In all the assays, except for duplex B, some high Cq values, ranging from 35.4 to 37.8, were observed for the ToBRFV signal in healthy seed samples. No assay displayed any reaction in healthy leaf and non-amplification controls (NAC 1, 2). Based on these results, the duplex assay using ToMMV set 3 targeting the MP region and the ToBRFV M&W assay (targeting the CP region) Duplex B, were selected for full validation.

### 2.2. Duplex Assay Validation

The selected duplex assay (Duplex B) was further evaluated and included in the validation process according to EPPO PM 7/98 (4) guidelines [38].

#### 2.2.1. Analytical Sensitivity

The LOD was determined using the 10-fold dilution series (leaves and seeds) obtained from ToMMV and ToBRFV mixed infected samples (Table 3). In leaves, the LODs were 10^−5^ for ToMMV and 10^−6^ for ToBRFV, whereas in seeds both viruses could be detected up to the 10^−5^ dilution (Figure 2a,b). In addition, the E (%) and R^2^ values were determined for each primer/probe set included in the duplex assay on the basis of the standard curves obtained in leaf and seed dilution series (Figure 3). Both sets had a similar efficiency in both matrices, with R^2^ close to the optimum value of 1 and a comparable efficiency above 95% (Table 3).

In addition, for each virus and matrix, the LOD was evaluated and the efficiency of the selected ToMMV and ToBRFV primers/probe sets was also calculated in a single assay (Table 3). In both ToBRFV and ToMMV single assays, the LOD was 10-fold lower than in the duplex assay (except for ToBRFV in leaves confirming a LOD of 10^−6^), and the efficiency was 1–2% lower in duplex than in single assays. However, the standard curves overlapped, sharing common slope values (Figure 4a,b).

#### 2.2.2. Analytical Specificity

The exclusivity was assessed by testing the target tobamovirus species by duplex assay at CREA and at NIB (Appendix A and Table 4). In general, no cross-reaction was observed, except for PaMMV and RMV; of these, only the cross-reaction with PaMMV may represent a threat for identification of ToBRFV and ToMMV, particularly in pepper, which is the chief host of PaMMV. RMV cross-reaction can be considered to be not important because it is restricted to ToBFRV and does not affect ToMMV. Further, its hosts are limited to *Brassica oleracea* and *Brassica chinensis*, and is not hosted by tomato and pepper.

The inclusivity of the duplex assay was assessed using different ToBRFV and ToMMV isolates (Appendix A and Table 5). In all cases, the identification of ToBRFV and ToMMV was successful. In addition, the *in silico* analysis versus all the full-genome sequences of ToMMV available in GenBank highlighted only up to two nucleotide polymorphisms in forward and/or reverse primers (Appendix A). However, these single point mutations were observed in a group of isolates including the DSMZ isolate PV-1267 that was shown to be successfully detected (Table 5).

#### 2.2.3. Selectivity

We showed that the duplex assay developed in this study can be used to detect and identify ToBRFV and ToMMV in leaf and seed matrices (Table 3). Furthermore, no relevant differences in Cq values were found (see Appendix A) when five tomato and six pepper cultivars spiked with ToMMV and ToBRFV were tested; in one tomato variety (pom-241 ‘sv5197’) both leaf and fruit matrices were tested (Appendix A).

#### 2.2.4. Repeatability and Reproducibility

The repeatability of the duplex assay was evaluated by analyzing three replicates of RNA samples containing various concentrations of ToBRFV or ToMMV in the same run. Repeatability within each sample was 100%, with standard deviation (SD) of the mean Cq values obtained always less than 1.5 Cq for the high, medium, and low quantities of target RNA (data not shown). Reproducibility also proved to be 100% (SD below 1.5 Cq; Table 6). Reproducibility was analyzed for two dilutions of a ToBRFV- and ToMMV-positive RNA samples, with medium and low target concentrations. The repeatability and reproducibility were assessed at both CREA and NIB, analyzing different target RNA samples and using different reagents (see Section 4). In each laboratory, different real-time RT-PCR runs, four (CREA) and six (NIB), were performed on different days. In addition, at NIB two different instruments were used (see Section 4). 

### 2.3. Use of the Validated Duplex Assay in Diagnosis of Seed Samples

To further assess the feasibility of the duplex assay developed and validated in this study for a rapid and sensitive detection of ToBRFV and ToMMV in seeds, a wide number of RNA samples extracted from seeds of different origins and previously assayed for ToBRFV were tested with the duplex assay.

A total of 106 tomato and pepper official seed samples from 2021 (RNA extracted from 62 tomato seed samples and from 44 pepper seed samples) from different origins (China, Italy, Brazil, India, Slovenia, Serbia) and previously analyzed for ToBRFV at CREA or at NIB, were included (Appendix A). In all these samples, the absence/presence of ToBRFV was confirmed with 100% concordant results; all three tomato and all five pepper seed samples, previously found to be ToBRFV positive, were also positively confirmed with the duplex ToBRFV and ToMMV assay. Regarding ToMMV, fifteen official seed samples resulted with Cq values in a range from 23.1 to 37.7 for ToMMV (Appendix A). Each of these 15 samples was further analyzed by generic tobamovirus nested PCR [37] and by Sanger sequencing of the nested PCR products; and the presence of ToMMV was clearly confirmed in five samples in which the Cq values of ToMMV ranged from 23.1 to 29.0. In samples with Cq values for ToMMV of 29.7 to 37.7, no nested PCR product or pure sequence was obtained, or the presence of another tobamovirus (ToMV) was confirmed (Appendix A). The origin of all seed samples in which the presence of ToBRFV or ToMMV was confirmed is China. ToBRFV was confirmed in seeds of four different pepper cultivars (‘Barbara’, ‘Galben superior’, ‘Pintea’, and ‘Stef’) and in seeds of three different tomato cultivars (‘Chiquita pot’, ‘Drops’, and ‘Silvia’), and the presence of ToMMV was confirmed in five different tomato cultivars (‘Amalia’, ‘Chiquita pot’, ‘Imola’, ‘Ruxandra’, ‘Sandybelle’).

## 3. Discussion

The development of diagnostic techniques that are as rapid, effective, and sensitive as possible is one of the most important issues in the controlling of plant viruses. In particular, regulated and/or emerging plant viruses may pose a major threat to most crops, and the information gained from rapid detection and identification is important for determining any measures needed to prevent the introduction of these pathogens and/or limit their spread. In view of this, it is essential to have reliable diagnostic tests. Recently, ToBRFV highlighted how easily viruses belonging to the *Tobamovirus* genus can pose a phytosanitary risk, especially for tomato and pepper crops. For this reason, prompt test performance studies (TPSs) organized under several national and transnational initiatives/European projects, in addition to the publication of the EPPO standard PM7/146 (1), have provided valid tools for the establishment of ToBRFV containment measures, which are also included in Commission Implementing Decision (EU) 2020/1191 and subsequent amendments.

Recently, the increasing number of reports about the presence of ToMMV increased the attention paid to this tobamovirus, which was included in the EPPO alert list in 2020. The experience gained with ToBRFV should provide the guidelines for managing the threat of this emerging virus. The lack of a method such as the real-time RT-PCR test for ToMMV is a major diagnostic gap, which this study aims to address.

In particular, in this study, a real-time RT-PCR test was developed for the detection and identification of ToMMV, which can also be used in a duplex assay with a primer/probe set of a test already validated and recommended in the EPPO standard PM7/146(1) for ToBRFV detection (i.e., M&W) [20].

The ToMMV assay developed in this study targeting the MP region confirmed that this region can be a suitable target for primer/probe set design. This region provides sufficient variability to distinguish between species of the genus *Tobamovirus*, which are known to have a high degree of homology [32]. In fact, the other pre-tested sets developed in this study and targeting the CP region reacted unspecifically with CGMMV, a tobamovirus that infects *Cucurbitaceae* plant species and a very limited number of *Solanaceae* species [39].

The ToBRFV and ToMMV duplex assay (Duplex B) included in the validation process according to [38] was evaluated in terms of analytical sensitivity and specificity, selectivity, repeatability, and reproducibility. The LODs obtained for both leaf (10^−6^/10^−5^) and seed (10^−5^) matrices and for both target viruses were in line with those of other real-time RT-PCR tests for the detection of tobamovirus species [20,40,41]. In addition, the performance criteria when testing dilution series (leaf or seeds samples) showed that there was no relevant reduction in LOD and/or primer efficiency (E) for the duplex assay compared to the single assays. The standard curves obtained in leaf and seed samples were similar, indicating equal slope and efficiency. The effectiveness of ToBRFV detection and identification in leaf and seed samples is not affected by either the ToBRFV M&W test alone or the duplex assay. This comparison is essential as the ToBRFV M&W test has already been validated (TPS under the H2020 Valitest and Euphresco 2019-A-327 projects) and is one of the two mandatory assays for ToBRFV detection in seeds (Commission Implementing Decision (EU) 2020/1191).

In addition, the duplex assay showed optimal analytical specificity on both inclusivity and exclusivity. Twenty-two non-target isolates of tobamovirus species were tested and the only relevant cross-reaction was obtained for paprika mild mottle virus—PaMMV (Table 5). Indeed, the high Cq values (29.3–35.0) for both ToMMV and ToBRFV (the latter in the single and duplex assay) may represent an unspecific reaction that should be considered because PaMMV shares common hosts (i.e., peppers) with them. Nevertheless, in silico analysis (data not shown) did not reveal sequence identity with both primer/probe sets that would warrant a cross-reaction. Moreover, PaMMV is reported to have sequence homology of about 64% with different ToMMV isolates [32]; in view of this, cross-reaction between the two targets does not seem likely to occur. Similar to PaMMV, cross-reactivity of RMV with ToBRFV in single and duplex assays is not expected due to sequence homology. Nevertheless, the samples of non-target tobamoviruses included in this study are from artificially inoculated test plants. A high virus concentration is expected in these samples; such a virus titer is hardly found in naturally infected leaf samples and not at all in seed samples, but further analysis and investigation are required.

The results obtained for inclusivity show that the duplex assay is able to detect all known ToBRFV and ToMMV isolates. In support of the inclusivity data (tested on few isolates from different sources), in silico analysis showed that the 14 different ToMMV whole genome sequences had very low variability, suggesting low selection pressure for this virus. In addition, point mutations were found in regions targeting forward and reverse primers (not the probe). However, these mutations occurred at the 5′/3′ ends of the primers, and in the DSMZ PV-1267 ToMMV isolate used throughout the development of this assay. This confirms that the proposed test is capable of identifying ToMMV and detecting all its isolates.

Results of repeatability and reproducibility, showing a standard deviation between tested samples of less than 1.5 Cq, confirm that this duplex assay can be used for the detection and identification of ToBRFV and ToMMV with a high degree of confidence. As these results were obtained independently by two laboratories (CREA and NIB) on different days and with different reagents, operators, and equipment, the associated validation data can be considered robust.

As mentioned above, ToMMV has been added in the EPPO alert list and is considered an emerging pathogen. As with ToBRFV, seed testing is important to prevent the spread of plant viruses, especially viruses that, like tobamoviruses, arise through mechanical/contact transmission. Recent reports of ToMMV occurrence, particularly in pepper seeds imported to Australia in 2020 [30] and in batches of tomato seeds imported by companies in Spain between 2019 and 2021 [31], confirm the importance of seed testing to ensure free movement without phytosanitary risks. Since all these reports used an endpoint RT-PCR, it is likely that the presence of ToMMV may be underestimated.

In view of the above, this study focused on the detection of ToBRFV and ToMMV on seeds.

One hundred and six tomato/pepper official seed samples previously tested for ToBRFV at CREA or NIB were assayed using the duplex assay developed in this study. Regarding ToBRFV, data showed 100% concordance with previous analysis.

The results of the 106 official seed samples showed that 15 samples taken during the inspection of seed lots imported from China gave some positive signals for ToMMV, where we could confirm the presence of ToMMV in five samples by Sanger sequencing of amplicons belonging to the region RdRp (Appendix A). In addition, one of these samples was tested and the presence of ToMMV was also confirmed by Sanger sequencing of the amplicon belonging to region CP (data not shown). Beyond the 106 official seed samples, no mixed ToMMV/ToBFRV infection was assessed. However, our results suggest that the spreading of ToMMV is underestimated, confirming the findings reported by [31], and that more accurate testing should be undertaken on seeds, especially those imported from other countries.

Samples confirmed positive for ToMMV by Sanger sequencing of amplicons yielded duplex real-time RT-PCR Cq values ranging from 23.1 to 29.0 (Appendix A). One sample with a Cq value of 29.7 is thought to have a mixture of ToMMV and ToMV, whereas all other nine samples that did not show ToMMV on Sanger sequencing had Cq values between 31.7 and 37.7. Due to the expected lower sensitivity of nested RT-PCR compared to the sensitivity of the real-time RT-PCR, and since the nested RT-PCR used is based on degenerate primer sets targeting the tobamovirus genus, and the species titers in mixed infections may be different, the presence of a low ToMMV titer in such samples is also possible, including hypothetical contaminations. Taking all results together, including the exclusivity tests, signals with high Cq values for ToMMV and ToBRFV (in both the single and duplex ToBRFV (M&W) assays) are signals that should be further evaluated.

In conclusion, based on the results obtained from this study, a duplex assay that allows simultaneous detection of ToBFRV and ToMMV was developed and validated. This detection test satisfies the requirements of validation according to international guidelines (i.e., EPPO PM 7/98 (4)) and can therefore be used as a rapid screening for the detection and identification of ToBRFV and ToMMV, even in cases in which a low concentration of these two viruses is expected, such as in seeds. A reliable and validated test represents an important diagnostic tool that allows prompt measures to be established to contrast a likely new threat for tomato and pepper crops. From the results of this work, it appears that the presence of ToMMV in seeds is underestimated and has not been comprehensively studied. This virus has been reported since 1992, and there are no known outbreaks of ToMMV or direct damage caused by this virus to susceptible crops; however, its incidence has been constantly increasing in recent years as indicated by the number of genome sequences added to GenBank in the last ten years. The control of ToMMV may be important, especially in terms of overall seed quality. The spreading of this virus and its free accumulation in seeds, despite not yet raising a phytosanitary alert, may represent a future risk for tomato and pepper. The worldwide outbreaks of ToBFRV from 2015 onward highlight how the official controls and phytosanitary measures were not prepared to contain its spread. In view of the above, a prompt evaluation of the presence of ToMMV together with ToBFRV and/or other tobamoviruses in seeds will strengthen phytosanitary strategies to prevent future outbreaks.

## 4. Material and Methods

### 4.1. Plant Material and RNA Extraction

ToMMV: tomato and *N. bethamiana*, ToBFRV: tomato and pepper used in this study are listed in Appendix A including: (i) isolates already characterized and included in CREA (Council for Agricultural Research and Economics, Research Centre for Plant Protection and Certification) and NIB (National Institute of Biology) collections; (ii) isolates purchased from DSMZ (German Collection of Microorganisms and Cell Cultures GmbH) or kindly provided by other institutions.

In addition, RNAs extracted from official samples of seeds from different seed lots originating from different countries and previously analyzed for ToBRFV at CREA or NIB (Appendix A) were also included. The total RNAs used for the assay set-up and validation were extracted according to the RNA extraction methods for leaves and seeds reported in the ToBRFV EPPO Standard PM 7/146 (1) [20]. Total RNA yield and quality were assessed by a NanoDrop ND1000 spectrophotometer (Thermo Fisher, Monza Italy), or real-time RT-PCR amplifying *nad5* [42] or *cytochrome oxidase* [43] were used as controls to evaluate the quality of the RNA in the extractions (data not shown). Buffer controls were included with all isolations (i.e., negative isolation control) to monitor potential contamination through the extraction procedures.

### 4.2. Primer Design and Reaction Set Up

The reference full genome sequence of ToMMV (NC_022230) was retrieved from GenBank and used as a template for TaqMan® probe and primer set design by Primer Express® Software Version 3.0 (Applied Biosystems, Foster City, CA, USA) with default parameters. In particular, two different regions coding for coat protein (CP) and movement protein (MP) were included in the design since they represent target regions for the two validated real-time RT-PCR tests available for ToBRFV detection [20]: the Menzel and Winter test (M&W) targeting a fragment from the end of the CP gene to the middle of 3-NTR) [20], and the International Seed Federation-International Seed Health Initiative for Vegetable Crops (ISHI-Veg) test, which is a duplex assay involving both CP and MP regions [20].

The designed TaqMan® probe and primer sets were synthesized by Eurofins genomics (Kolhn, Germany) and labeled in HEX and/or Texas Red according to the correct match with the fluorophore used in ToBRFV detection tests based on real-time RT-PCR (i.e., M&W, FAM-labeled, and ISHI-Veg, FAM/HEX-labeled). Real-time RT-PCR reactions were performed using the CFX96 optical reaction module with a C1000 Thermalcycler (Bio-Rad, Milan, Italy). Preliminary reactions were performed including two negative (healthy leaf and seed samples), one ToMMV-positive (ToMMV isolate PV-1267 from DSMZ), one ToBRFV-positive (ToBRFV isolate MR50—leaf), and one negative amplification control, in two technical repetitions. These samples were tested at different primers and probe concentrations and annealing temperature ranging from 55 to 62 °C (data not shown). Once the optimized amplification conditions were determined, reactions were carried out in 10 µL reaction volumes containing 5 µL of 2× Mastermix, 0.25 µL of 40× RT Enzyme (both from TaqMan® RNA-to-Ct™ 1-Step Kit, Thermo Fisher, Milan, Italy), 0.3 µM of each primer, 0.2 µM of labeled TaqMan® probe, and 1 µL of template RNA. The optimized one step real-time RT-PCR cycling conditions included a RT step at 48 °C for 30 min, an initial denaturation step at 95 °C for 10 min followed by 40 cycles of denaturation, and annealing/elongation for 15 s at 95 °C and 1 min at 60 °C, respectively. Analyses were performed using the BIORAD CFX Maestro v2.2. For assessing exclusivity, each ToMMV primer/probe set was tested including the following non-target isolates from DSMZ collection: TMV (PV-0137), ToMV, tobacco mild green mosaic virus—TMGMV, bell-pepper mosaic virus—BPeMV, pepper mild mottle virus—PMMoV, and cucumber green mottle mosaic virus—CGMMV (PV-0375), retrieved from DSZM collection (Appendix A). Further, ToMMV primer/probe sets were evaluated in duplex/triplex assay with the corresponding ToBRFV test (M&W or ISHI-Veg), according to targeting region and/or labeling. The duplex/triplex assays were evaluated on a sample set composed of healthy leaf/seed controls, leaf/seed samples infected by ToBRFV (ascertained to be in single infection), ToMMV leaf samples, and a mixed ToMMV and ToBRFV infected sample (prepared by mixing the same amount (0.05 g) of freeze-dried material of ToBRFV MR50 isolate and ToMMV PV1276 isolate. Based on the general quality assessment, i.e., amplification curve patterns, value of quantification cycle (Cq), concordance of relevant controls, amplification background signals, and performance in the duplex/triplex assay with ToBRFV tests taking into account the different primer/probe set targeting region, the best ToMMV primer/probe set was selected to be further validated as a duplex assay at CREA.

Selected duplex (ToMMV and ToBRFV M&W) real-time RT-PCR was also tested by NIB according to the following mixture reaction (final volume of 10 µL): 2 µL of sample RNA, 0.3 µM of each primer, 0.2 µM of each labeled TaqMan® probe] using the AgPath-ID™ One-Step RT-qPCR mix (Thermo Fisher, Monza, Italy). Real-time RT-PCR was carried out in 384-well plates (Applied Biosystems, Foster city, CA, USA), and the reactions were run on an QuantStudio™ 7 Pro System or ViiA7™ System sequence detection (Applied Biosystems, Foster city, CA, USA). One-step real-time RT-PCR cycling conditions included a RT step at 48 °C for 10 min, an initial denaturation step at 95 °C for 10 min followed by 45 cycles of denaturation and annealing/elongation for 15 s at 95 °C and 1 min at 60 °C, respectively.

The Design & Analysis Software v2.5.1 (Applied Biosystems Foster city, CA, USA) was used for fluorescence acquisition and determination of Cq values. Before analysis, ROX^TM^ reference dye was excluded to prevent its interference with Texas Red dye, and the baseline was set manually. The fluorescence threshold was also set manually, i.e., to a level that was above the baseline and sufficiently low to be within the exponential increase region of the amplification curve. If no exponential amplification curve was produced, a sample was considered negative. If an exponential amplification curve was produced, the Cq values were determined. In addition, especially for ToBRFV, a Cq threshold value was determined as needed according to the report in [20].

### 4.3. Duplex Assay Validation

The ToBRFV and ToMMV duplex assay was validated according to [38]. To determine all the criteria included in the validation process, a sample set summarized in Appendix A was used.

#### 4.3.1. Analytical Sensitivity, Limit of Detections and Efficiencies

Eight ten-fold serial dilutions using both leaf and seed matrices were prepared. To assess the analytical sensitivity in leaf samples, artificially mixed infected samples were created by mixing the same amount of freeze-dried material (0.05 g each) derived from leaves infected with ToBRFV (isolate MR50) and ToMMV (isolate PV—1276 from DSMZ). The sample was homogenized with PO_4_ buffer (pH 7.2) and 100 µL was serially diluted in 900 µL of healthy crude extract (tomato plant), to obtain a total eight 10-fold serial dilutions. Then, for each aliquot of the dilution series, total RNA was extracted by a RNeasy Plant Mini Kit (Qiagen, Hilden, Germany), adding 100 µL to 380 µL of RLT buffer (included in the kit), and following manufactures’ instructions.

To assess the analytical sensitivity in seed samples, two ToBRFV infected seeds (CREA collection) and 5 mg of freeze-dried leaf material infected by ToMMV (PV-1276 from DSMZ) were spiked in 998 tomato healthy seeds. The material was transferred in a grinding bag (BIOREBA XL), soaked in GH+ buffer, as reported in [20], for 1 h at room temperature, and homogenized with an Interscience BagMixer (position 4) for 90 s. Eight ten-fold serial dilutions were prepared (100 µL of homogenized infected sample in 900 µL of homogenized healthy tomato seeds), and instructions reported in [20] were followed.

All the aliquots from the serial dilution for each matrix were assayed by the duplex assay for ToMMV and ToBRFV.

For both matrices (leaf and seed) the limit of detection (LOD) was determined, the amplification efficiencies (*E*) calculated, and all the results obtained compared. In detail, standard curves were obtained by plotting mean Cq values of the ten-fold serial dilutions (leaves and seeds) versus minus the logarithm of the RNA dilution factors. The Cq values and the following equation were used to determine the efficiency (*E*) of each primer/probe set with the slope of a linear regression model; the linear correlation coefficient (R^2^) was also reported.
$$E\left(\%\right)=\left(10^{1-slope}-1\right)\times 100$$

The same dilution series were tested and the efficiency also calculated for cases where selected the ToMMV primer/probe set and ToBRFV assay were used as single assay.

#### 4.3.2. Analytical Specificity

A total of 13 different tobamovirus species were assayed at CREA and NIB as a relevant non-target (Appendix A) to assess the exclusivity parameter; for some tobamovirus species (TMV, ToMV) commonly present in tomato and/or pepper, different isolates were included. The inclusivity was assessed using a total of six ToBRFV and ToMMV isolates, 3 each (Table 5) according to different geographic origins and hosts. In addition, the inclusivity results were further widened by in silico analysis, by aligning (Clustal W supported by MEGA X software [44]) the duplex assay primer/probe sets versus all the full genome sequences of ToMMV and ToBRFV retrieved from GenBank (Appendix A).

#### 4.3.3. Selectivity

To determine whether variations of the sample material could affect the test performance, different tomato and pepper cultivars and matrices (leaf, seed, and fruit) were tested (Appendix A) In detail, 30 µL of homogenate derived from mixed infected samples (prepared as reported in Section 4.3.1) was added to 70 µL of healthy crude extracts of different tomato/pepper varieties/matrices. The total RNA was extracted by a RNeasy Plant Mini Kit (Qiagen) (see Section 4.3.1) and tested using the duplex assay developed in this study.

#### 4.3.4. Repeatability and Reproducibility

These performance parameters were tested by performing technical replicates and repeating the assay in two laboratories (i.e., CREA and NIB), with different operators, on different days, and with different instruments; deviations due to the use of different reagents were also checked.

## 5. Conclusions

The ToBRFV and ToMMV real-time RT-PCR duplex assay developed in this study was found to be reliable for use in routine analyses, according to international validation guidelines. Moreover, this aspect appears to be of fundamental importance for seed testing, where the requirements for a sensitive and reliable diagnostic tool are crucial given the phytosanitary risk posed by the free movement of seeds worldwide. Since testing of seeds for ToBRFV is mandatory in most countries, the development of a duplex assay for two tobamoviruses as presented in this study may provide an effective diagnostic tool for testing imported seeds, especially for those countries where ToMMV must also be tested (e.g., Australia). A rapid and sensitive diagnostic test is essential for the introduction of containment measures to protect tomato and pepper crops.

## Figures and Tables

**Figure 1 plants-11-00489-f001:**
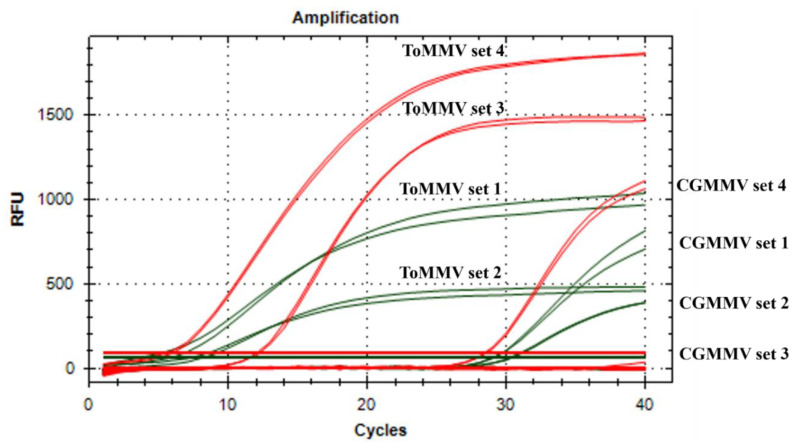
Amplification curves obtained from testing each ToMMV primer/probe set versus ToMMV and non-target tobamovirus species (ToBRFV, TMV, ToMV, TGMV, BPeMV, PMMoV, CGMMV). Cross-reaction of some primers/probe sets was observed only with CGMMV. (Probe labeling: green for HEX, red for Texas Red).

**Figure 2 plants-11-00489-f002:**
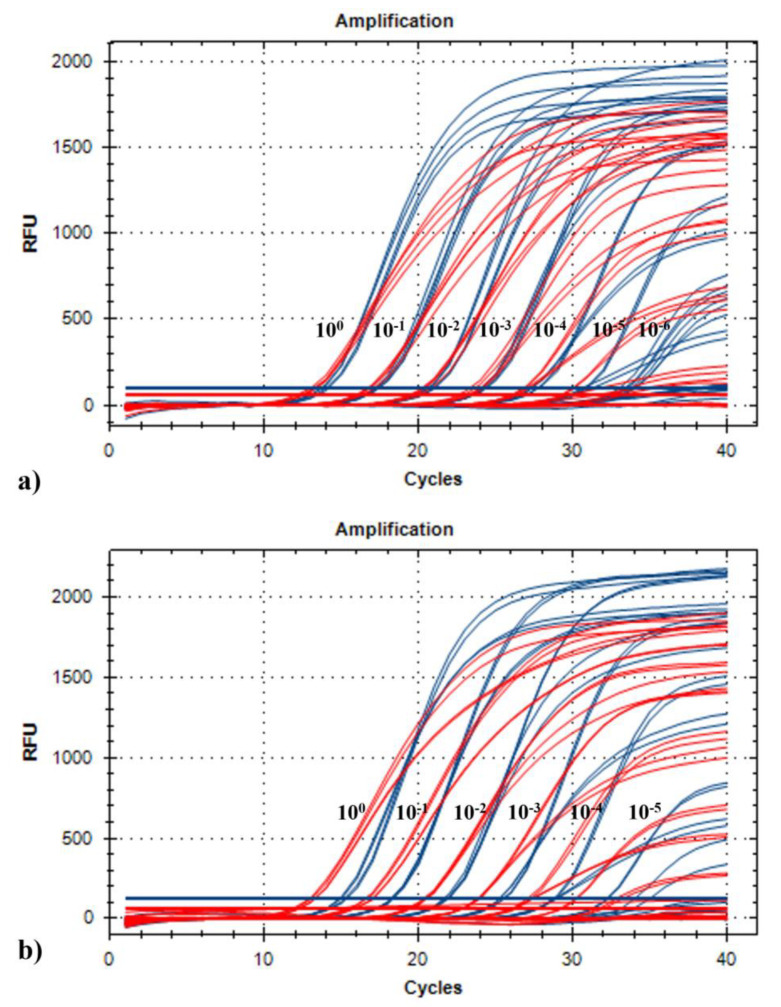
Amplification curves obtained from testing the 10-fold dilution series of leaf material (panel (**a**)) and seed material (panel (**b**)) in mixed infection ToMMV and ToBRFV, with the selected duplex assay: Duplex B: ToMMV set 3 (probe labeled with Texas Red, curves in red) and ToBRFV M&W (probe labeled with FAM, curves in blue)).

**Figure 3 plants-11-00489-f003:**
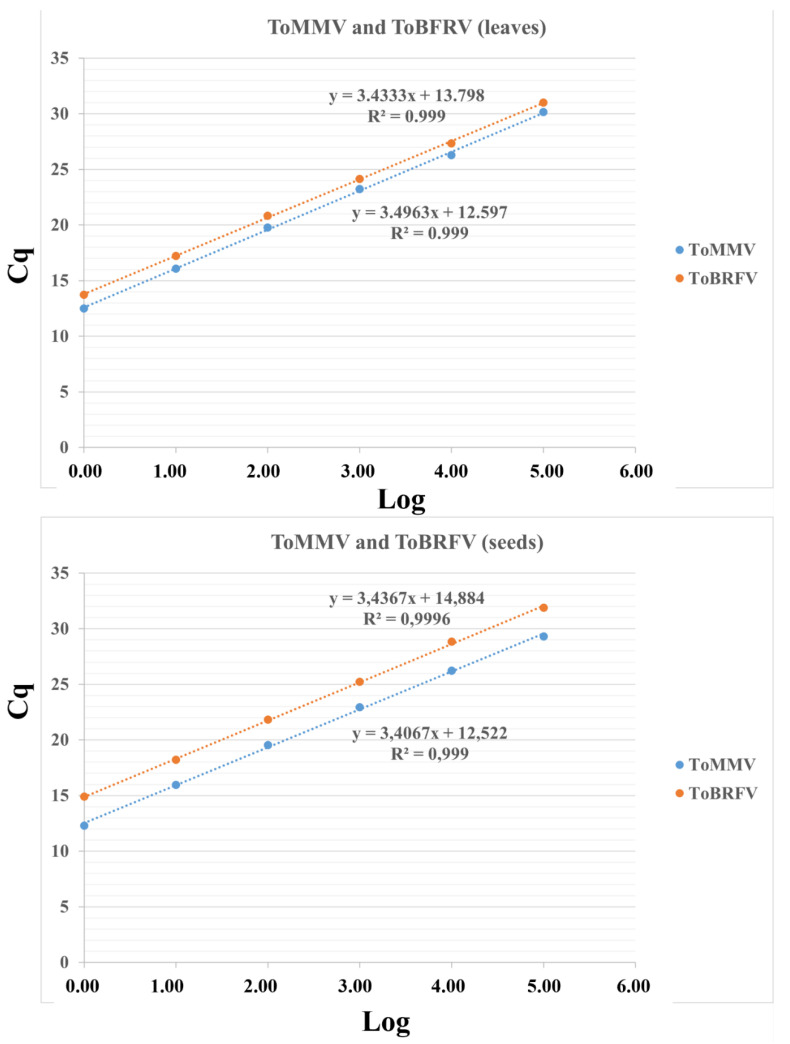
Standard curves of the ToBRFV and ToMMV TaqMan® duplex assay in leaf and seed samples obtained from ten-fold serial dilutions. The quantification cycle (Cq) value is plotted against the log of RNA ten-fold serial dilutions. It was possible to adjust a straight line. The value of the slope reported in the straight-line equation allowed for the estimation of the efficiency of the reaction. R^2^ is a measure of the adjustment to a linear model.

**Figure 4 plants-11-00489-f004:**
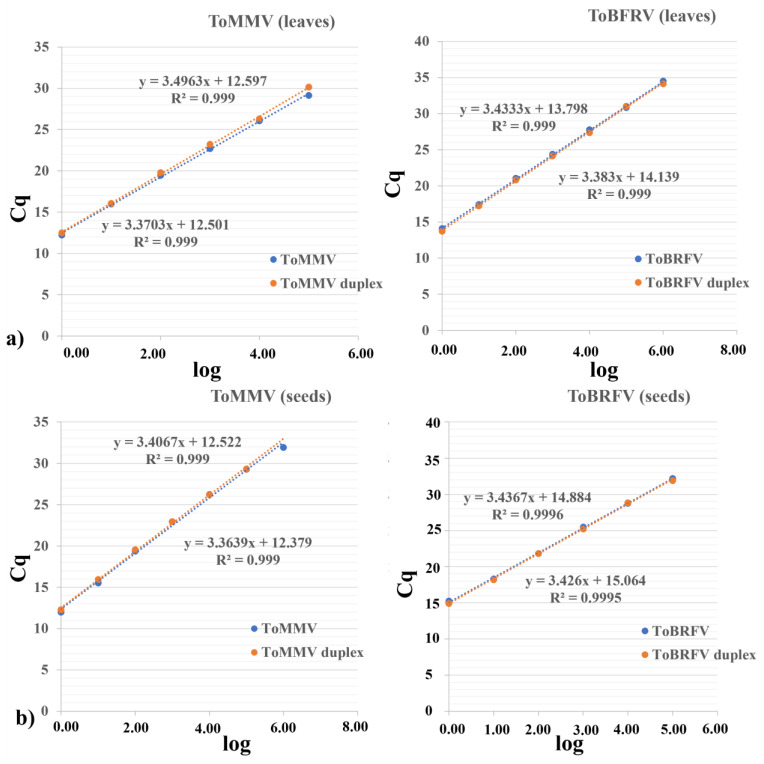
TaqMan® assay standard curves obtained in the single and in the duplex assay (for ToMMV on the right and ToBRFV on the left), in leaf (panel (**a**)) and seed (panel (**b**)) samples using ten-fold serial dilutions. The quantification cycle (Cq) value is plotted against the log of RNA ten-fold serial dilutions. It was possible to adjust a straight line. The value of the slope reported in the straight-line equation allowed for the estimation of the efficiency of the reaction. R^2^ is a measure of the adjustment to a linear model.

**Table 1 plants-11-00489-t001:** Primers/probe sets designed on ToMMV reference sequence (NC_022230) and targeting two genome regions (MP and CP).

Set	Primer/Probe ID	5′-3′ Sequence	Labeling	Genome Position (nt)	Coding Region
1	ToMMV MWAT Fw	GGAACAGGTTTCTACAACCAGAG		6124–6146	CP
	ToMMV MWAT Rv	CGTACGCACCACGTGTATTT		6235–6254	
	ToMMV MWAT Pr1	TTCTGCGCCAGCGTCCTAAGTAAT	HEX/BHQ1	6180–6203	
2	ToMMV MWAT Fw	GGAACAGGTTTCTACAACCAGAG		6124–6146	CP
	ToMMV MWAT Rv	CGTACGCACCACGTGTATTT		6235–6254	
	ToMMV MWAT Pr2	GGCCTGGACTTCTGCGCCA	HEX/BHQ1	6171–6189	
3	ToMMV CataAT Fw	CAGCATCTGCTTGGTCGATAA		5173–5193	MP
	ToMMV CataAT Rv	GGAACGATCTTAAACTGGAACCT		5255–5277	
	ToMMV CataAT Pr	AATGCAAAGAGCGGATGAAGCGAC	Texas Red/BHQ2	5197–5220	
4	ToMMV CSPAT Fw	GGAACAGGTTTCTACAACCAGAG		6124–6146	CP
	ToMMV CSPAT Rv	CGTACGCACCACGTGTATTT		6235–6254	
	ToMMV CSPAT Pr	TTCTGCGCCAGCGTCCTAAGTAAT	Texas Red/BHQ2	6180–6203	

All four primers/probe sets were tested versus some other tobamovirus species (TMV, ToMV, TGMV, BPeMV, PMMoV, CGMMV). None of the sets cross-react with TMV, ToMV, TGMV, BPeMV, and PMMoV. However, all three primers/probe sets targeting CP resulted in cross-reaction with CGMMV, whereas no signal was observed for the set targeting the MP region (set 3) (Figure 1).

**Table 2 plants-11-00489-t002:** Quantification cycle (Cq) values obtained for each ToMMV and ToBRFV duplex/triplex assay analyzing the same sample set. (HL: healthy leaf; HS: healthy seeds; IL: infected leaf; IS: infected seeds; NAC: negative amplification control; NA: no exponential amplification curve). Results considered as positive are marked in black bold.

	Duplex A		Duplex B		Triplex A			Triplex B		
	ToBFRV M&W	ToMMV Set 1	ToBRFV M&W	ToMMV Set 3	ToBRFV ISHI-Veg		ToMMV Set 3	ToBRFV ISHI-Veg		ToMMV Set 4
	CP-FAM	CP-HEX	CP-FAM	MP-TexasRed	CP-FAM	MP-HEX	MP-TexasRed	CP-FAM	MP-HEX	CP-TexasRed
HL	NA	NA	NA	NA	NA	NA	NA	NA	NA	NA
HS	37.8 *	NA	NA	NA	36.0 *	36.5 *	NA	35.4 *	36.1 *	NA
ToBRFV IL	**16.2**	NA	**17.5**	NA	**18.2**	**18.4**	NA	**18.7**	**18.8**	NA
ToBRFV IS	**27.9**	NA	**25.8**	NA	**28.4**	**28.2**	NA	**28.5**	**28.1**	NA
ToMMV IL	NA	**9.2**	NA	**12.3**	NA	NA	**13.6**	NA	NA	**8.8**
Mixed sample IL	**15.2**	**10.1**	**16.6**	**12.3**	**18.6**	**19.2**	**14.4**	**18.2**	**18.8**	**9.6**
NAC1	NA	NA	NA	NA	NA	NA	NA	NA	NA	NA
NAC2	NA	NA	NA	NA	NA	NA	NA	NA	NA	NA

* Cq above the Cq threshold considered as negative (no clear exponential amplification curve).

**Table 3 plants-11-00489-t003:** Quantification cycle values obtained for the 10-fold dilution series (in both leaf and seed) tested for ToBRFV (in single or duplex assay) and for ToMMV (in single or duplex assay). In addition, for each assay the E (%) and R^2^ values were determined and are reported in the table.

	Leaves				Seeds			
	ToBRFVs	ToBRFVd	ToMMVs	ToMMVd	ToBRFVs	ToBRFVd	ToMMVs	ToMMVd
**10^0^**	**14.1**	**13.7**	**12.2**	**12.5**	**15.2**	**14.9**	**12.0**	**12.3**
**10^−1^**	**17.4**	**17.2**	**16.0**	**16.1**	**18.3**	**18.2**	**15.5**	**15.9**
**10^−2^**	**21.1**	**20.8**	**19.5**	**19.8**	**21.8**	**21.8**	**19.4**	**19.5**
**10^−3^**	**24.4**	**24.2**	**22.7**	**23.2**	**25.4**	**25.2**	**22.9**	**23.0**
**10^−4^**	**27.8**	**27.4**	**26.1**	**26.3**	**28.8**	**28.8**	**26.3**	**26.2**
**10^−5^**	**30.8**	**31.0**	**29.1**	**30.2**	**32.2**	**31.9**	**29.3**	**29.3**
**10^−6^**	**34.5**	**34.1**	**32.8**	NA	**34.5**	NA	**31.9**	NA
**10^−7^**	38.8 *	NA	NA	NA	NA	NA	NA	NA
**10^−8^**	39.4 *	NA	NA	NA	NA	NA	NA	NA
**R^2^**	**0.999**	**0.999**	**0.999**	**0.999**	**0.999**	**0.999**	**0.999**	**0.999**
**E**	**97.63**	**95.67**	**98.03**	**95.20**	**95.83**	**95.42**	**97.33**	**96.57**

s—single assay; d—duplex assay; NA—no exponential amplification curve; * Cq above the Cq threshold and considered as negative (no clear exponential amplification curve).

**Table 4 plants-11-00489-t004:** Results of testing different isolates of non-target tobamoviruses with the duplex ToBRFV and ToMMV assay and the single ToBRFV M&W assay.

			NIB			CREA		
			Single Assay	Duplex		Single Assay	Duplex	
Virus	Collection	ID	ToBRFV M&W	ToBRFV (Cq)	ToMMV (Cq)	ToBRFV M&W	ToBRFV (Cq)	ToMMV (Cq)
BPeMV	DSMZ	BN-4708	nt	nt	nt	NA	NA	NA
			nt	nt	nt	NA	NA	NA
			nt	nt	nt	NA	NA	NA
CGMMV	NIB	NIB V 271	NA	NA	NA	nt	nt	nt
			NA	NA	NA	nt	nt	nt
			NA	NA	NA	nt	nt	nt
CGMMV	NIB	NIB V 320	NA	NA	NA	nt	nt	nt
			NA	NA	NA	nt	nt	nt
			NA	NA	NA	nt	nt	nt
CGMMV			nt	nt	nt	NA	NA	NA
	DSMZ	PV-0375	nt	nt	nt	NA	NA	NA
			nt	nt	nt	NA	NA	NA
ObPV	DSMZ	PV-1176	38.4 *	NA	NA	nt	nt	nt
			NA	NA	NA	nt	nt	nt
			NA	NA	NA	nt	nt	nt
ORSV	DSMZ	PV-1048	NA	NA	NA	nt	nt	nt
			NA	NA	NA	nt	nt	nt
			NA	NA	NA	nt	nt	nt
PaMMV	DSMZ	PV-0606 (2020) **	**33.9**	**33.5**	**31.5**	nt	nt	nt
			**33.8**	**33.5**	**31.4**	nt	nt	nt
			**33.4**	**33.7**	**31.4**	nt	nt	nt
PaMMV	DSMZ	PV-0606 (2021) **	**33.8**	**34.2**	**29.6**	nt	nt	nt
			**35.0**	**34.3**	**29.5**	nt	nt	nt
			**34.0**	**33.7**	**29.3**	nt	nt	nt
PMMoV	CREA	CREA-552	nt	nt	nt	NA	NA	NA
			nt	nt	nt	NA	NA	NA
			nt	nt	nt	NA	NA	NA
PMMoV	DSMZ	PV-0165	nt	nt	nt	NA	NA	NA
			nt	nt	nt	NA	NA	NA
			nt	nt	nt	NA	NA	NA
RMV	DSMZ	PV-0145	**33.9**	**36.9**	NA	nt	nt	nt
			**34.4**	**35.2**	NA	nt	nt	nt
			**34.8**	**37.4**	NA	nt	nt	nt
SFBV	DSMZ	PV-1058	NA	NA	NA	nt	nt	nt
			NA	38.2 *	NA	nt	nt	nt
			NA	NA	NA	nt	nt	nt
SHMV	DSMZ	PV-0156	NA	NA	NA	nt	nt	nt
			NA	NA	NA	nt	nt	nt
			NA	NA	NA	nt	nt	nt
ToMV	NIB	NIB V 036	NA	NA	NA	nt	nt	nt
			NA	NA	NA	nt	nt	nt
			NA	NA	NA	nt	nt	nt
ToMV	NIB	NIB V 049	NA	NA	NA	nt	nt	nt
			NA	NA	NA	nt	nt	nt
			NA	NA	NA	nt	nt	nt
ToMV	NIB	NIB V 072	NA	NA	NA	nt	nt	nt
			NA	NA	NA	nt	nt	nt
			NA	NA	NA	nt	nt	nt
ToMV	NIB	NIB V 104	NA	NA	NA	nt	nt	nt
			NA	NA	NA	nt	nt	nt
			NA	NA	NA	nt	nt	nt
ToMV	DSMZ	PV-0141	nt	nt	nt	NA	NA	NA
			nt	nt	nt	NA	NA	NA
			nt	nt	nt	NA	NA	NA
TMGMV	DSMZ	PV-0124	NA	NA	NA	NA	NA	NA
			NA	NA	NA	NA	NA	NA
			NA	NA	NA	NA	NA	NA
TMV	NIB	NIB V 037	NA	NA	NA	nt	nt	nt
			NA	NA	NA	nt	nt	nt
			NA	NA	NA	nt	nt	nt
TMV	DSMZ	PV-1252	nt	nt	nt	NA	NA	NA
			nt	nt	nt	NA	NA	NA
			nt	nt	nt	NA	NA	NA
TMV	DSMZ	PV-0137	NA	NA	NA	NA	NA	NA
			38.5*	NA	NA	NA	NA	NA
			NA	NA	NA	NA	NA	NA
TMV	DSMZ	PV-0943	NA	NA	NA	NA	NA	NA
			NA	NA	NA	NA	NA	NA
			NA	NA	NA	NA	NA	NA
YMoV	DSMZ	PV-0527	NA	NA	NA	NA	NA	NA
			NA	NA	NA	NA	NA	NA
			NA	NA	NA	NA	NA	NA

NA—no exponential amplification curve; nt-not tested; * Cq above the Cq threshold and considered as negative (no clear exponential amplification curve); ** two batches of isolate were tested (one purchased from DSMZ in 2020 and one in 2021).

**Table 5 plants-11-00489-t005:** Results of testing different isolates of ToMMV and ToBRFV with the duplex ToBRFV and ToMMV assay and the single ToBRFV M&W assay.

			Single	Duplex	
Virus	Collection	ID	ToBRFV M&W (Cq)	ToBRFV M&W (Cq)	ToMMV (Cq)
ToMMV	DSMZ, DE	PV-1267	36.9 *	NA	**8.4**
			38.9 *	NA	**8.0**
			37.6 *	NA	**8.1**
ToMMV	IBMCP; SP	S1	nt	NA	**12.7**
			nt	NA	**12.8**
			nt	NA	**12.9**
ToMMV	IBMCP; SP	S2	nt	NA	**15.9**
			nt	NA	**15.7**
			nt	NA	**15.7**
ToBRFV	CREA, IT	MR50 (100,000 × dilution)	**25.5** **25.6** **25.3**	**25.7** **26.1** **26.0**	NA NA NA
ToBRFV	Volcani center; IS	S21	nt	**6.7**	NA
			nt	**6.8**	NA
			nt	**5.1**	NA
ToBRFV	Volcani center; S	S22	nt	**6.8**	NA
			nt	**4.7**	NA
			nt	**4.6**	NA

NA—no exponential amplification curve; nt-not tested; * Cq above the Cq threshold and considered as negative (no clear exponential amplification curve). DE: Germany; IS: Israel; IT: Italy; SP: Spain.

**Table 6 plants-11-00489-t006:** Results of testing medium and low concentration of targets and negative amplification controls with the duplex ToBRFV and ToMMV assay at NIB and at CREA on different days using different instruments, operators, and reagents.

		ToBRFV (Cq)	ToMMV (Cq)	ToBRFV (Cq)	ToMMV (Cq)	ToBRFV (Cq)	ToMMV (Cq)	ToBRFV (Cq)	ToMMV (Cq)	ToBRFV (Cq)	ToMMV (Cq)	ToBRFV (Cq)	ToMMV (Cq)
	Run ID Samples	1		2		3		4		5		6	
NIB	ToBRFV PC1	**25.6**	NA	**26.7**	NA	**24.7**	NA	**25.6**	NA	**25.3**	NA	**25.5**	NA
	ToBRFV PC2	**31.8**	NA	**32.5**	NA	**31.2**	NA	**30.3**	NA	**31.0**	NA	**30.4**	NA
	ToMMV PC1	NA	**26.2**	NA	**26.6**	NA	**25.7**	NA	**27.8**	NA	**27.2**	NA	**27.3**
	ToMMV PC2	NA	**32.9**	NA	**33.0**	NA	**32.9**	NA	**35.4**	NA	**34.2**	NA	**35.6**
	NAC1	NA	NA	NA	NA	NA	NA	NA	NA	NA	NA	NA	NA
	NAC2	NA	NA	NA	NA	NA	NA	NA	NA	NA	NA	NA	NA
CREA	ToBRFV PC1	**23.4**	NA	**23.6**	NA	**23.3**	NA	**23.4**	NA	nt	nt	nt	nt
	ToBRFV PC2	**30.8**	NA	**30.8**	NA	**31.1**	NA	**31.0**	NA	nt	nt	nt	nt
	ToMMV PC1	NA	**24.4**	NA	**24.9**	NA	**24.8**	NA	**25.2**	nt	nt	nt	nt
	ToMMV PC2	NA	**31.4**	NA	**31.2**	NA	**31.2**	NA	**32.0**	nt	nt	nt	nt
	NAC1	NA	NA	NA	NA	NA	NA	NA	NA	nt	nt	nt	nt
	NAC2	NA	NA	NA	NA	NA	NA	NA	NA	nt	nt	nt	nt

NA—no exponential amplification curve; nt-not tested; NAC—negative amplification control; PC1—positive control with medium target (ToBRFV/ToMMV) concentration; PC2—positive control with low target (ToBRFV/ToMMV) concentration.

## Data Availability

Not applicable.

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
