# Peer review of "Development and Validation of a One-Step Reverse Transcription Real-Time PCR Assay for Simultaneous Detection and Identification of Tomato Mottle Mosaic Virus and Tomato Brown Rugose Fruit Virus"

_plants, 2022, doi:10.3390/plants11040489_

Round 1

Reviewer 1 Report

This study developed a one-step RT-PCR approach to detect and identify ToMMV and ToBRFV in tomato leaves and seeds. The data showed here are very convincing. I would just suggest that the authors should explain what are CP and MP regions at the beginning. e.g., Line 108.   

Author Response

Reviewer 1:

First of all we would like to really thanks the comments and the effort made by the reviewer aimed to improve the manuscript.

This study developed a one-step RT-PCR approach to detect and identify ToMMV and ToBRFV in tomato leaves and seeds. The data showed here are very convincing. I would just suggest that the authors should explain what are CP and MP regions at the beginning. e.g., Line 108.   

R: the text was amended accordingly

Reviewer 2 Report

Dear Authors,

I have great honor and opportunity to review paper entitled: “Development and validation of a one-step reverse transcription  real-time PCR assay for simultaneous detection and identification of tomato mottle mosaic virus and tomato brown rugose fruit virus” which is considered for publication in Plants journal. Article is generally good written but has some inappropriate approach to wrote scientific article in Plants journal rules. The wrote style is presented in a way to show methodological approach. In this context the Plants journal is not exactly suitable for this type of article. It will be better suitable for other MDPI journals like Methods and Protocols or Viruses. However, I believe with a lot of work it is possible to rearrange/rewrite article in a way suited for Plants journal. My specific comments I present in a form of list below.

  1. General comment

In general Authors must point by point check Plants journal publication rules and modify the article according to that. Currently in many points article is not prepared according journal publication rules. For example any of 48 references on reference list is not prepared according journal rules. Authors need to remember that all journal names must be abbreviation in ISO standard.

  1. Specific comments:
  2. Introduction section:

This section must introduce the scientific knowledge and problem and next present in precise scientific way the aim of the study and if this is possible also hypothesis according Plants journal. Currently this section has 41 references but no clearly presented scientific aim and/or hypothesis of authors research

  1. Results section

Less problematic

Table 1 must be move to separate way currently the name of table and most of table is on 2 pages which looks little strange. It also possible to move the table into the supplement.

Table 2 must be reconstructed in a way that all table cells could be readable. Moreover the scientifically significant results -positive results should be marked with “*” not negative one. I suggest the positive in bold with “*”Moreover the NAC1 and NAC2 must be characterized in description of table.

Table 3 Moreover, the scientifically significant results -positive results should be marked with “*” not negative one

Table 4 Moreover, the scientifically significant results -positive results should be marked with “*” not negative one. I also suggest to move Table 4 to the supplementary data.

More problematic:

Many of Figures are data from basic software of used devices and in this context they do not look like in high quality. I strongly suggest to make Figures in other programs specialized for presentation of this type study. This program occurs in case of Figure 1 and Figure 2. Moreover authors in Figure 2 need to add information which curve blue or red is ToBRFV or ToMMV or maybe chart A  or B is for ToBRFV or ToMMV?.

Figure 3 is too low quality and too small

Discussion is generally too short in the context of length of results, this makes the article not equally balanced.

Sincerely,

Author Response

Reviewer 2:

First of all we would like to really thanks the comments and the effort made by the reviewer aimed to improve the manuscript. We replied to all comments included point by point

Dear Authors,

I have great honor and opportunity to review paper entitled: “Development and validation of a one-step reverse transcription  real-time PCR assay for simultaneous detection and identification of tomato mottle mosaic virus and tomato brown rugose fruit virus” which is considered for publication in Plants journal. Article is generally good written but has some inappropriate approach to wrote scientific article in Plants journal rules. The wrote style is presented in a way to show methodological approach. In this context the Plants journal is not exactly suitable for this type of article. It will be better suitable for other MDPI journals like Methods and Protocols or Viruses. However, I believe with a lot of work it is possible to rearrange/rewrite article in a way suited for Plants journal.

R: The manuscript has been rearranged to be suitable for Plants journal rules. In our opinion the manuscript contents fulfill the topics of the special issue of the journal "Tobamoviruses and Interacting Viruses in Modern Agriculture" in which the manuscript has been planned to be published. We believe that our article will be of interest to readers of that special issue. As mentioned on the website, the goal of that special issue is also to have articles about effective disease control strategies. Control of viral diseases relies on preventing their spread. Accurate diagnosis of tobamoviruses is a prerequisite for management of the two tobamoviruses studied here.

 My specific comments I present in a form of list below.

  1. General comment

In general Authors must point by point check Plants journal publication rules and modify the article according to that. Currently in many points article is not prepared according journal publication rules. For example any of 48 references on reference list is not prepared according journal rules. Authors need to remember that all journal names must be abbreviation in ISO standard.

R: the references were amended accordingly to Plants guidelines

  1. Specific comments:
  2. Introduction section:

This section must introduce the scientific knowledge and problem and next present in precise scientific way the aim of the study and if this is possible also hypothesis according Plants journal. Currently this section has 41 references but no clearly presented scientific aim and/or hypothesis of authors research

R: we amended the text accordingly. In this work we aimed to develop and validate a method, the introduction helped us to establish the state of the art and the background to justify the reason why this method is need.

  1. Results section

Less problematic

Table 1 must be move to separate way currently the name of table and most of table is on 2 pages which looks little strange. It also possible to move the table into the supplement.

Table 2 must be reconstructed in a way that all table cells could be readable. Moreover the scientifically significant results -positive results should be marked with “*” not negative one. I suggest the positive in bold with “*”Moreover the NAC1 and NAC2 must be characterized in description of table.

Table 3 Moreover, the scientifically significant results -positive results should be marked with “*” not negative one

Table 4 Moreover, the scientifically significant results -positive results should be marked with “*” not negative one. I also suggest to move Table 4 to the supplementary data.

R: we amended all the tables accordingly to the reviewer’s comment to make it clear. Positive results were marked in bold with cell background in grey. The “*” mark is used (as explained at the end of all tables) for cases where Cq was above the Cq threshold or no clear amplification curve were detected. The Cq value in addition were typed in light grey. Further, in our opinion is important to maintain table 4 since the cross-reactions could really affect the reliability of a method.

More problematic:

Many of Figures are data from basic software of used devices and in this context they do not look like in high quality. I strongly suggest to make Figures in other programs specialized for presentation of this type study. This program occurs in case of Figure 1 and Figure 2. Moreover authors in Figure 2 need to add information which curve blue or red is ToBRFV or ToMMV or maybe chart A  or B is for ToBRFV or ToMMV?.

Figure 3 is too low quality and too small

R: the images were amended and loaded in highest quality. In addition, the figures will be sent to the journal in a zip file separately in the highest quality.

Discussion is generally too short in the context of length of results, this makes the article not equally balanced.

R: In our opinion the length of the Discussion section, including the conclusion, is good enough. We reviewed the discussion section again and found that all topics relevant to the study are included in the discussion. All relevant results are summarized and explained, and the results of our study are compared with the results of previous work. In addition, any shortcomings that occurred during the course of the work are mentioned. The Results part is longer than the Discussion part, but full of tables and figures.

Reviewer 3 Report

Dear authors

I made my comments in the Ms. The Ms is of great interest for enforcement laboratories dealing with the detection of ToBRFV in tomato and pepper seeds and plants. However, there are some issues that need clarification.

Author Response

Reviewer 3

First of all we would like to really thanks the comments and the effort made by the reviewer aimed to improve the manuscript. We replied to all comments included point by point

Comment 1: page 2

Do you want to mean the development or resistance genes or resistant varieties? I understand that resistance genes existing in related species may be introgressed into tomato varities. Therefore, the gene is not developed, but the variety may be developed.

R: The sentence was amended according to reviewer’s opinion

Comment 2 page 3

I suggest to use "contaminated" instead of "infected". Contamination happens when the pathogen is present but infection is associated with the disease. Therefore, a positive test would be expected for infected plants.

R: We agree, the sentence was amended according to reviewer’s opinion

Comment 3 page 3

From which plant species - tomato or pepper? This is not clear also in material and methods. The authors should clarify because observing figure 1 and 2, there is a clear reduction in the RFU level. Was it due to a matrix effect?

R: It was used tomato plant material. The material and methods section was amended according to that.  The RFU reduction is not due to matrix effect but by the use of the ToMMV in duplex assay with ToBRFV primer/probe set.

Comment 4 page 3

Which quencher fluophore was used?

R: the table was amended accordingly. For the probes labeled in HEX BHQ1 quencher was used, whereas for probes with TexasRed we used BHQ2

Comment 5 page 4

I would suggest the authors to include the quencher and other molecules used. For instance, this probe is short which would benefit from the presence of MGB. Did you try MGB?

R: the table was amended accordingly. We did not try the MGB probes but it is our intention to try them in next planned activities.

Comment 6 page 4

Figures have legends rather than titles. Please, move this text to the bottom of the figure.

R: We amended all the figures accordingly

Comment 7 page 4

delete "versus" and use "with"

R: The text was amended

Comment 8 page 4:

Why there is only one curve for each ToMMV set and CGMMV set? In material and methods, you refered that each sample was used in 2 technical replicates.

R: The figure 1 previously loaded derived by a preliminary assay performed in single technical replicate and using a 25 mL mix. The correct figure was now loaded using the reaction mixture of 10 mL and primers/probe concentration set up in this study. The results, in terms of Cq values and cross-reactions observed were the same. The only difference was the RFU values obtaining using the 25 mL or 10 mL reaction mixtures, due to the use of reduced amounts of probes included in mixture. We amended the results section accordingly.

Comment 9 page 4

Set 3 gave RFU above 3000. Why did it give only 2000 in figure 2?

R: As reported in the previous reply, the figure 1 was replaced. Now the RFU values of set 3 in Figure 1 and Figure 2 are the same. (see reply to comment 8 above)

Comment 10 page 5

How repeatable was this result? How do you justify that the same method gives Ct 37.8 in duplex A and NA in duplex B? How many times did you repeat this experiment? Taking the information from MM, you have use only 1 seed sample which is not clear if it was tomato or pepper. Could you better explain this result?

R: The Ct value was indeed of 37.8 but a no clear exponential curve could be observed, and thus it was considered negative. This result did not surprise us since in the ToBFRV assay detection, especially in seeds, often could produce a no clear exponential curve, independently from the phytosanitary status of the samples, and the number of the repetition. A recent TPS report highlighted this issue for both ToBFRV tests used in seeds (M&W and ISHI-Veg - https://zenodo.org/record/5776210#.Ybc4Ur3MKUl). In addition, this was a preliminary experiment done to verify the best matching of using the different ToMMV sets with the ISHI-VEG or Menzel&Winter assay. The best duplex assay was then included in the validation process. We used tomato seeds. The material and method section was amended accordingly.

Comment 11 page 7

again, it is visible a difference in the RFU level between leaves and seeds (blue curves). Is there a matrix effect? The authors are asked to clarify this issue.

R: Please see the replies to comment 8 and 9. The blue curves as reported in the figure legend correspond to the ToBRFV assay whereas the red to ToMMV, for leaf (panel A) and seeds (panel B) matrices.

Comment 12 page 7

Taking this first sentence, readers are tempted to look for the efficiency values in the graphs.

R: we amended the text of figure legend. Reporting only the standards curves

Comment 13 page 7

I do not agree. This is the log of the dilution applied to the extract.

R: the text was amended.

Comment 14 page 8

LOD is not clearly stated as there is no mention to the concentration in 10E0 extract.

R: as reported in the EPPO standard 7/98 for virus, viroids and phytoplasmas the exact concentration is not known, therefore the maximum dilution of the sample giving a positive result is considered as LOD.

Comment 15 page 8

this is not intuitive as there is not value given for the efficiency. Please include in the figures the value of the reaction efficiency.

R: the values are reported in the table 3 (in the sentence, we added reference to Table 3). The table was amended to highlight the values of Efficiency. In our opinion including all the values in the text or in a figure could create confusion.

Comment 16 page 8

the layout of the tables are not friendly. I had to make columns by hand to better visualize. Please change this format.

R: the table layout was amended accordingly

Comment 17 page 9

I suggest to introduce the "exclusivity" word to be in line with teh wording "specificity inclusivity".

R: we agree and amended the text

Comment 18 page 9

Which criteria were applied to determine that the observed cross-reaction was not important? I suggest the authors to give two or three criteria that may be applied during the analysis of the amplification curves.

R: we amended the text to clear this point. We had only two cross-reactions. We include a sentence to justify why we do not consider relevant the cross-reaction with RMV.

Comment 19 page 12

not available!!! (Regarding figure S1)

R: It was in the zip file along with the supplementary tables

Comment 20 page 12

The number of isolate is very reduced. It would be adviceble the authors to present a justification for such situation. Why didn't you test isolates from Mexico or USA? I suggest you to add China origin in the section.

R: we agree about this comment. The obtaining of ToMMV isolates represented a big issue. Actually, we included ToMMV isolates from Spain and DSMZ (USA) origin. We tried to ask in different laboratories, just few of them had material that sent us. To reinforce this lacking, we performed in-silico analysis, including 14 ToMMV isolates, which full genome is available, and from different regions and matrices. According to EPPO standard PM7/98 in lacking of isolates, the in-silico analysis can overcome the impossibility to test a great number of isolates.

Comment 21 page 12:

I suggest authors to cleary say that only 3 cultivars of tomato and 5 varities of pepper tested positive. At least for tomato, 3 varieties is a very reduced number. Therefore, an explanation is needed to justify that 3 varieties is enough to waranty selectivity.

R: According to the comment above,    matrix effect has been further investigated on some other tomato and pepper varieties . The data are included as supplementary material (Supplementary Figure S2) and the Supplementary table S1 is amended. No effect was observed. In addition, in our opinion, among different varieties of the same species a great variability could not be observed, as it could be determined just from few loci. However, when official seed samples were tested, we detected ToBRFV or ToMMV in seeds of seven different tomato cultivars and in seeds of four different pepper cultivars. Seeds of all four pepper cultivars (Barbara, Galben superior, Pintea, Stef) and seeds of three tomato cultivars (Chiquita pot, Drops, Silvia) were infected with ToBRFV, while the presence of ToMMV was confirmed in five different tomato cultivars (Amalia, Chiquita pot, Imola, Ruxandra, Sandybelle). This explanation is added in a text, but since we are aware that this is not selectivity, this is added in the section 2.3.

Comment 22 page 13

Could you check if this is 10E-6? if 10E-5 gave Cq=25.6, 10E-6 should be 28.9 approximately, but not 31.8.

R: we amended the text to clear this point. We prepared positive controls by mixing RNA extracted from the target virus. The starting material for the positive control labelled 1 and the positive control labelled 2 were not the same. To avoid confusing readers, the text below the table has been simplified.

Comment 23 page 13

If you stated LOD at 10E-5 and 10E-6, why did you dilute until 10E-8? This is confusing and I fear that the LOD needs to be expressed in other way. the dilution level is not, metrologically speaking, the appropriate way.

R: we amended the text to clear this point (see above)

Comment 24 page 17

this point should be separated from i) and ii) because it refers to other table (S2). The whole paragraph is not very clear and I suggest to rewrite. For instance, I understood that the matrices in table S1 are the ones used to preserve the isolates. This is not clear to the reader. Table S2 does not have any reference to RNAs

R: the text was amended. The Tables S2 reported the samples that were already tested in official analysis for ToBRFV. To better specify that are RNA sample it was included in the title of the Supplementary Table S2.

Comment 25 page 17

This sentence needs to be revised as it has two verbs. After the reference 47 include a ". These RNAs were used...".

R: the sentence was reworded accordingly to reviewer’s comment.

Comment 26 page 17

tomato or pepper? Why did you did not use more samples and the two species? How can you discard the effect of the plant genome composition if only one seed sample was used?

R: we specify that tomato plant material was used. These were only preliminary analysis performed to have a preliminary evaluation only about the behavior of the primers/probe sets. Then the assay was included in the validation process including further controls and samples to exclude this possibility.

Comment 27 page 17

Figure 1 has less amplification curves than expected. ToMMV+(x2) + ToBRFV+(x2) + C-(x2)= 6 for each reaction conditions. Please, see my comments on figure 1.

R: Please to see replies to comments 9 and 11.

Comment 28 page 17

µL rather than µl.

R: we amended the text

Comment 29 page 17

why this range? When was used 43 and when was used 0.4? In the analytical sensitivity assays, how much RNA was used in the first and the 8th dilution? Taking the range 0.4-43, it was not possible to perform 8-ten fold serial dilutions.

R: we amended the text; it was a refuse. In the preliminary analysis done to evaluate the different sets we used a range of 50 – 500 ng samples to be in the range of the manufacturer’s recommendation (up to 1 mg).

Comment 30 page 18.

Why did you other master mix? Have you evidence on the performance of TaqMan® RNA-to-Ct™ 1-Step Kit, Thermo Fisher,Milan, Italy, when in the duplex assay?

R: all the validation process performed by CREA was done using TaqMan® RNA-to-Ct™ 1-Step Kit. At NIB the Agpath mastermix was used. The performances were similar. We included a second master mix to increase the robustness of the method.

Comment 31 page 19

Following the same rational, I suggest you to use "exclusivity" for the tests done with non-target species.

  1. we agree and amend the text accordingly

Comment 32 page 19

according to table S1, three origins were tested for ToBRFV and 2 for ToMMV. I suggest to be more precise in this sentence and not to use "several".

R: we amended the text accordingly

Comment 33 page 19

I do not agree with this statement. All samples present in table S2 were commercial samples, with an unknown status concerning the infection with ToBRFV orToMMV. To test the matrix effect on the detetion of viruses, all those varieties should prior be infected.

R: we agree, therefore this part is deleted. See reply to comment 21. Further analyses were performed, and the text was amended accordingly.

Round 2

Reviewer 1 Report

The manuscript can be accepted in the current form from my side. 

Reviewer 2 Report

Dear Authors,

Thank very much for improvements of manuscript. I recomend publication.

Sincerely,

Author Response

We would like to thanks the reviewr for the effort done in revising the manuscript. It allowed us to improve our manuscript.

Reviewer 3 Report

Dear authors

The Ms improved and just few details need to be corrected. Comments 25 and 32 were not considered although the answer was as they have been amended.
